# MiR-302d inhibits TGFB-induced EMT and promotes MET in primary human RPE cells

**Xiaonan Hu, Maximilian Binter, Karsten Hufendiek, Jan Tode, Carsten Framme, Heiko Fuchs** *

Institute of Ophthalmology, University Eye Hospital, Hannover Medical School, Hannover, Germany

* fuchs.heiko@mh-hannover.de

**Data Availability Statement:** All relevant data are within the article and its Supporting Information files.

**Funding:** The authors received no specific funding for this work.

## Abstract

### Purpose

Transforming growth factor-beta (TGFB)-mediated epithelial-mesenchymal transition (EMT) plays a crucial role in the pathogenesis of retinal fibrosis, which is one of the leading causes of impaired vision. Current approaches to treating retinal fibrosis focus, among other things, on inhibiting the TGFB signaling pathway. Transient expression of microRNAs (miR-NAs) is one way to inhibit the TGFB pathway post-transcriptionally. Our previous study identified the miRNA miR-302d as a regulator of multiple TGFB-related genes in ARPE-19 cells. To further explore its effect on primary cells, the effect of miR-302d on TGFB-induced EMT in primary human retinal pigment epithelium (hRPE) was investigated *in vitro*.

### Methods

hRPE cells were extracted from patients receiving enucleation. Transfection of hRPE cells with miR-302d was performed before or after TGFB1 stimulation. Live-cell imaging, immunocytochemistry staining, Western blot, and ELISA assays were utilized to identify the alterations of cellular morphology and EMT-related factors expressions in hRPE cells.

### Results

hRPE cells underwent EMT by TGFB1 exposure. The transfection of miR-302d inhibited the transition with decreased production of mesenchymal markers and increased epithelial factors. Meanwhile, the phosphorylation of SMAD2 activated by TGFB1 was suppressed. Moreover, miR-302d expression promoted TGFB1-induced fibroblast-like hRPE cells to revert towards an epithelial stage. As confirmed by ELISA, miR-302d reduced TGFB receptor 2 (TGFBR2) and vascular endothelial growth factor A (VEGFA) levels 48 hours after transfection.

### Conclusions

The protective effect of miR-302d might be a promising approach for ameliorating retinal fibrosis and neovascularization. MiR-302d suppresses TGFB-induced EMT in hRPE cells via downregulation of TGFBR2, even reversing the process. Furthermore, miR-302d reduces the constitutive secretion of VEGFA from hRPE cells.

**Competing interests:** The authors have declared that no competing interests exist.

## Introduction

Fibrosis is a repairing process in response to tissue damage [1]. It is characterized by an excessive deposition and remodeling of the extracellular matrix (ECM) and infiltration of myofibroblasts [2]. Retinal fibrosis manifests as the end-stage of many ocular diseases, including proliferative vitreoretinopathy (PVR), wet age-related macular degeneration (wAMD), proliferative diabetic retinopathy (PDR), retinopathy of prematurity (ROP), and choroidal neovascularization (CNV) [3]. The major pathological event is the retinal pigment epithelium (RPE)'s epithelial-mesenchymal transition (EMT), resulting in transdifferentiation into myofibroblasts [4]. Subsequently, these epithelium-originated fibroblast-like cells proliferate and migrate over the retina, promoting ECM production and wound contraction, resulting in distorted tissue structure and malfunction [1, 5].

During EMT, RPE cells undergo several changes. They experience the loss of polarity, disruption of intercellular tight junctions (TJs), characterized by the loss of Zonula Occludens 1 (ZO-1) and Occludin, and following "cadherin switch" from the expression of E-cadherin to N-cadherin [4, 6–11]. Remodeling the cellular cytoskeleton into stress fibers gives the RPE cell layer migratory and contractile properties [4, 7, 9, 10]. The release of cytokines and matrix metalloproteinases (MMPs) causes damage and degradation of the basement membrane, allowing mesenchymal RPE cells to migrate away and invade [4, 8, 12].

The transforming growth factor-beta (TGFB) family participates in numerous physiological and pathological conditions [13, 14]. It is also a potent inducer of EMT in retinal fibrosis [4, 13–15]. Through binding to cell surface receptors (TGFBR1 and TGFBR2), TGFB activates its canonical SMAD signaling pathway and plenty of non-canonical signaling pathways, resulting in increased activation, expression, and nuclear translocation of transcriptional factors such as Snail1, Snail2, Twist1, and Zeb1, which are critical regulators of EMT [12, 13, 16]. It is widely verified that inhibiting TGFB signaling pathways at different levels can efficiently suppress cancer and fibrosis [14]. Current approaches include antagonizing TGFB ligands, blocking TGFB receptors, and inhibiting TGFB signaling-related proteins [6, 14]. As small RNAs' involvement in eye diseases has been more and more elucidated, mediating TGFB function in EMT of RPE cells by applying small RNAs has become a relatively novel method [6, 14, 17].

MicroRNAs (miRNAs) are small, non-coding, single-stranded RNAs that bind to the 3'UTR of their target genes by imperfect pairing and inhibit their protein synthesis [18]. Depending on their binding affinity, miRNAs can also destabilize the mRNA of their target genes [19]. More than 2000 miRNAs are reported to involve in cell proliferation, differentiation, signaling, and multiple diseases [17]. Emerging evidence shows that some miRNAs play a positive role in the pathogenesis of several ocular diseases, such as PVR, AMD, and PDR [6, 17, 20]. Our previous study demonstrated that transient overexpression of miR-302d or miR-93 could inhibit TGFB-induced EMT and vascular endothelial growth factor A (VEGFA) secretion in ARPE-19 cells and trigger mesenchymal-epithelial transition (MET) in dedifferentiated ARPE-19 cells [6]. Though the ARPE-19 cell line is a widely used alternative for primary human RPE (hRPE) cells, it may lack some critical properties of native hRPE cells [21–23].

Therefore, our study aims to identify the protective effect of miR-302d on EMT in primary hRPE cells.

## Methods

### Cell isolation and culture

Primary hRPE cells were obtained from two female patients who received enucleations suffering from an early stage of painful Phthisis bulbi. Patients with previous eye disorders involving

the retina were excluded. The eyeballs were packed in ROTI®Cell Hanks' BSS (HBSS, Carl Roth #9117.1) on ice and processed to harvest RPE cells within 1–2 hours after surgery. The procedures were carried out following the tenets of the Declaration of Helsinki, with the consent of the ethics committee of the Hannover Medical School. An informed written agreement was obtained from all subjects. The eyeball was briefly dipped in 70% ethanol for disinfection. Next, it was cut perpendicular to the equator in a petri dish filled with HBSS using a stereomicroscope. The anterior segment, vitreous body, and retina were removed. The RPE sheet was peeled off with forceps and then digested with 1 ml TrypLE™ Express Enzyme (Gibco #12604–021) on a thermo-shaker for 30 min, 37˚C, and 600 rpm. The detached RPE cells were collected by centrifugation and then resuspended and cultured in a 6-well plate with Minimum Essential Medium Eagle media (MEM, Sigma-Aldrich #M8042) supplemented with 10% fetal bovine serum (FBS, Pan-Biotech #P40-39500), 1% GlutaMAX™ (Gibco #3505–061), and 1% Penicillin-Streptomycin (Pen-Strep, Gibco #15140–122). The same RPE sheet was digested two more times for 30 min each, and the RPE cells were transferred to different wells. Sequential digestion was performed to avoid over digestion of already detached cells, resulting in poor attachment. When the cells were confluent, they were examined with a phase-contrast and brightfield microscope (Leica DMi1) regarding their epithelial properties and degree of pigmentation. Wells showing RPE cells with similar morphology were pooled in passage 1 with 400 μl TrypLE™, and cells of passages 4–5 were used for experiments. It should be mentioned that RPE cells seeded at low density changed to a mesenchymal state. Therefore, the cells were seeded at a density of 50% to maintain their epithelial state, i.e., the cells of one confluent 6-well were divided into two 6-wells.

## RNA isolation and quantitative polymerase chain reaction (qPCR)

$6 \times 10^4$ RPE cells were seeded in each well of a 6-well plate until they reached confluence. Total RNA was isolated with TRI-Reagent (Sigma #T9424) based on the manufacturer's protocol with some minor modifications. Briefly, 5PRIME Phase Lock Gel–Heavy tube (Quantabio #2302830) was used for phase separation, and 1 ml instead of 0.5 ml of Isopropanol was used for miRNA/RNA precipitation.

Complementary DNA (cDNA) was synthesized from 1 μg total RNA using customized specific primers and the TaqMan® MicroRNA RT Kit (Thermo Fisher #4366596) in a thermal cycler under the condition of 16˚C for 30 minutes, 42˚C for 30 minutes, 85˚C for 5 minutes, and 4˚C on hold. MiRNA qPCR was performed on a CFX96 Touch™ Real-Time PCR Cycler (Bio-Rad) under the following condition: 10 minutes of enzyme activation followed by 40 cycles of denaturing (95˚C for 15 minutes) and extension (60˚C for 1 minute). FAM fluorescence was recorded, and the result was analyzed using the CFX™ Maestro Software. Samples with Cq values > 35 were classified as not available (N/A). All samples were measured in four technical and three biological replicates.

## miRNA transfection and TGFB treatment

$2 \times 10^4$ primary hRPE cells were seeded in each well of a 24-well plate with a complete medium to reach 70–80% confluency one day before the transfection. Before transfection, the medium was changed to MEM with 2% FBS without Pen-Strep. The cells were transfected with 1X phosphate-buffered saline (PBS, Biowest #L0615) as the mock group, 5 pmol miRNA mimic negative control (NC, Ambion #4464058), 5 pmol miR-302d (Ambion #4464066) or 10 μM SB431542 (Selleckchem #S1067), a TGFBR1 inhibitor, using Lipofectamine® RNAiMAX (Invitrogen #13778075) according to the manufacturer's protocol.

For TGFB treatment, 10–20 ng/ml recombinant human TGFB1 (Peprotech #100–21) was added to the medium, depending on the cell confluency.

## Immunocytochemistry (ICC) staining and uptake of green fluorescent microbeads

hRPE cells were seeded on 13 mm diameter microscope cover glasses (Glaswarenfabrik Karl Hecht #41001113) in a 24-well plate. 0.5 μl green fluorescent microbeads (SPHERO™ #FL-2052-2) were added to the medium for 96 h to verify the phagocytic property of the RPE cells. Before fixation, the RPE cells were washed three times with 500 μl PBS to remove unincorporated microbeads.

For the other treatments, cells were washed twice with 500 μl PBS, then fixed with ROTI®-Histofix 4% (Carl Roth #P087.4) for 30 min at room temperature (RT). The fixed cells were permeabilized and blocked with a blocking solution containing 0.1% Triton™X-100 (Sigma-Aldrich #X-100), 2% goat serum (Millipore #S26-100ML), 1% bovine serum albumin (BSA, Sigma-Aldrich #A2153), and 0.05% Tween®20 (Sigma-Aldrich #P9416) in PBS for 30 min at RT. Cells were then incubated with primary antibodies of RPE65 Mouse mAb (Invitrogen #MA1-16578), Vimentin Rabbit mAb (VIM, Cell Signaling #5741S), ZO-1 Rabbit mAb (Cell Signaling #13663S), Fibronectin 1 Rabbit mAb (FN1, Cell Signaling #26836S), or alpha-smooth muscle actin Rabbit mAb (αSMA, ACTA2, Cell Signaling #19245S), which were diluted 1:1000 in the blocking solution, overnight at 4˚C. Following incubation with 1:1000 diluted Alexa Fluor™ 488 or Alexa Fluor™ 546 goat anti-mouse or anti-rabbit secondary antibody (Invitrogen #A11029, #A11008, #A11035) and 1:500 Alexa Fluor™ 488- or rhodamine-conjugated phalloidin (Invitrogen #A12379, #R415) in PBST (0.1% Tween®20 in PBS) at RT for two hours, the coverslips were mounted upside down on a microscope slide with ROTI®-Mount FluorCare DAPI (Carl Roth #HP20.1). Images were taken with Observer Z.1 microscope (Carl Zeiss) and ZEN-Blue analysis software (Carl Zeiss). Additionally, the Apotome2 device was used to record the uptake of fluorescent beads.

## Western blot

hRPE cells were lysed with 1X Laemmli Sample Buffer (Bio-Rad #1610737) containing a 1X protease inhibitor cocktail (PI, Cell signaling #5871S). The lysates were mixed with 2-Mercaptoethanol (Sigma-Aldrich #60-24-2), denatured at 95˚C for 5 min, and stored at -20˚C. The samples were loaded on gels made by TGX Stain-Free™ Fast-Cast™ Acrylamide Kit (Bio-Rad #1610181). After 100–120 V electrophoresis, the resolved proteins were transferred to an ethanol-activated Mini-size LF PVDF membrane (Bio-Rad #10026934) with Trans-Blot®Turbo™ Transfer System at 1.3 mA, 25 V for 7 min or 1.3 mA, 25 V for 10 min for high molecular weight proteins. The PVDF membrane was blocked with 5% milk powder (Carl Roth #T145.2) in 1X Tris-buffered saline (TBS) at RT for one hour, followed by incubation with 1:1000 diluted primary FN1 antibody (Cell Signaling #26836S) overnight at 4˚C. After incubation with 1:1000 diluted goat anti-rabbit secondary antibody StarBright™ Blue 700 (Bio-Rad #12004162) at RT for one hour, the fluorescent signal was detected with ChemiDoc MP Imaging System (Bio-Rad) at a measurement wavelength of 660–720 nm. ImageLab 6 software (Bio-Rad) was used for normalization and quantification.

## ELISA assay

Before cell lysis, the media from hRPE cells were stored after centrifugation to detect extracellular VEGFA. hRPE cells were then lysed with 1X radioimmunoprecipitation assay buffer

(RIPA, Abcam #ab156034) containing 1X PI. The lysates were centrifuged at 14000 RCF for 15 min, and the supernatants were stored at -80˚C for future analysis of TGFBR2.

Pierce™ BCA Protein Assay Kit (Thermo Scientific #23225) was used to normalize total protein concentrations according to the manufacturer's protocol. TGFBR2 level in the lysate supernatants and VEGFA level in the media were measured with Human TGF-beta RII Duo-Set ELISA Kit (R&D Systems #DY241) and Human VEGF Quantikine ELISA Kit (R&D Systems #DVE00) respectively according to the manufacturer's instructions. The color change was measured using a Tecan Spark® M10 plate reader. All samples were measured in four technical replicates and three biological triplicates.

### Live-cell imaging and analysis

Live-cell imaging was performed using BioTek® Lionheart™ FX automated microscope. The settings were performed as previously described with minor modifications [6]. Briefly, the 24-well plate was placed in the humidity chamber with 5% $CO_2$ and 37˚C. The program was set to phase-contrast channel and 4X PL FL objective. Default autofocus and autoexposure images were taken at 30 min intervals.

The analysis of images was performed with Gen5 Image prime 3.05 software. The images were first processed to flatten the background. The background was set to "Light" and "Image smoothing strength" to 20 "Cycles of 3x3 average filter". The "Rolling Ball diameter" was set to 20 μm and "Priority" to "Fine results". For advanced analysis of processed pictures, the "Threshold" was configured to 1000, "Rolling Ball diameter" to 100 μm, "Image smoothing strength" to 3 "Cycles of 3x3 average filter", and the background was evaluated on 5% of lowest pixels. The object size was set between 20–200 μm with the options of "Include primary edge objects", "Analyze entire image", "Fill holes in masks" and "Split touching objects" selected. Metrics of "object length" and "object width" were calculated.

### Wound-healing assay

hRPE cells transfected with miR-302d or NC were seeded in a 24-well plate. A scratch was performed with a 200-μl pipette tip when cells reached confluency. After 2 PBS rinses, serum-free medium with or without 10 ng/ml TGFB1 was supplemented. The scratch was imaged under Live-cell imaging for 3 days. The wound area was measured using Gen5 Image prime 3.05 software. The wound confluency was calculated as the percentage of the original wound area covered by migrating cells using Microsoft Excel 2010 and GraphPad Prism 9.

### Statistical analysis

All experiments were performed in three independent replicates, and the data were analyzed using Microsoft Excel 2010 and GraphPad Prism 9. Differences were evaluated by unpaired two-tailed t-test, one-way ANOVA with Tukey's multiple comparisons test or two-way ANOVA with Fisher's LSD test as the post hoc test. A $p$ value < 0.05 was considered significantly different.

## Results

### Identification of primary hRPE cells

In order to identify hRPE cells, some specific characteristics were examined. ICC staining revealed abundant expression of RPE65, a native marker of RPE cells, in extracted cells. Moreover, fluorescent microbeads were added into the medium for four days to test the phagocytic property of RPE cells. The green fluorescent microbeads were visible within the cytoplasm,

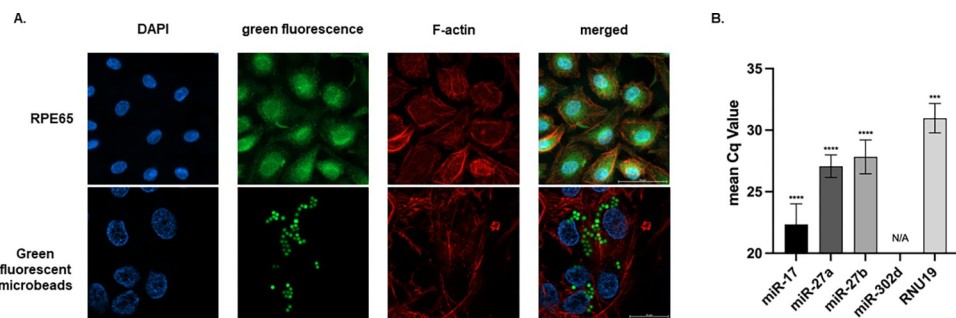

**Fig 1. Identification of hRPE cells and miR-302d expression.** (A) hRPE cells were immunostained for RPE65 and green fluorescent microbeads. The scale bar represents 50 μm for RPE65 and 20 μm for microbeads. (B) Mean Cq values of miR-17, miR-27a, miR-27b, miR-302d, and RNU19 of TaqMan® MicroRNA assay. Cq values > 35 were classified as not available. One-sample t-tests (n = 3) were performed to detect differences between mean Cq values and 35. The error bars represent SD. N/A, not available, ***$p < 0.001$, ****$p < 0.0001$.

which was confirmed by the coexistence with Filamentous actin (F-actin). The findings validated that primary hRPE cells were successfully extracted (Fig 1A).

In addition, a TaqMan® MicroRNA assay was performed to determine whether hRPE cells endogenously express miR-302d. MiR-17, miR-27a, and miR-27b were used as controls, as their expressions were reported in RPE cells, and RNU19 was used as the housekeeping gene [24, 25]. According to the results, hRPE cells do not endogenously express miR-302d (Fig 1B).

## miR-302d inhibits TGFB-induced fibroblastic morphological change in hRPE cells

Primary hRPE cells were exposed to TGFB1 24 h after transfection with mock, NC, or miR-302d to verify the inhibitory function of miR-302d on TGFB-induced mesenchymal alterations. Phase-contrast live-cell imaging was performed for the next 120 h to document the morphological alteration. At the end of the recording, hRPE cells transfected with mock or NC appeared elongated and spindle-like. In contrast, cells transfected with miR-302d maintained a similar appearance to cells not exposed to TGFB1 (Fig 2A).

In addition, the mean length-width ratio of hRPE cells was measured using Gen5 software to quantify the morphology change. After 5 d treatment of TGFB1, the mean length-width ratio of hRPE cells transfected with mock or NC increased from 2.6 and 2.5 to 2.9 and 3.1. In contrast, the ratio of miR-302d transfected cells increased subtly from 2.4 to 2.5. A significant difference between control groups and miR-302d could be observed already after 24 h ($p = 0.0046$, Fig 2B).

## miR-302d inhibits TGFB-induced EMT in hRPE cells

Besides live-cell imaging, ICC staining was performed to localize the distributions of EMT-related factors. In the absence of TGFB1, VIM localized around the nucleus, while TGFB1 caused wide spreading of VIM in the cytoplasm in control groups. Phalloidin staining revealed the stress fiber formation in response to TGFB1. F-actin was rearranged from a radial to a parallel distribution in mock or NC transfected cells. The expression of ZO-1, an epithelial marker in intercellular TJs, decreased after 4 d exposure to TGFB1, whereas the deposition of mesenchymal factor αSMA increased. In contrast to the control groups, miR-302d transfected cells retained an epithelial phenotype similar to those without TGFB1 treatment (Fig 3). In agreement with the above data, immunoblot and immunostaining confirmed that miR-302d could

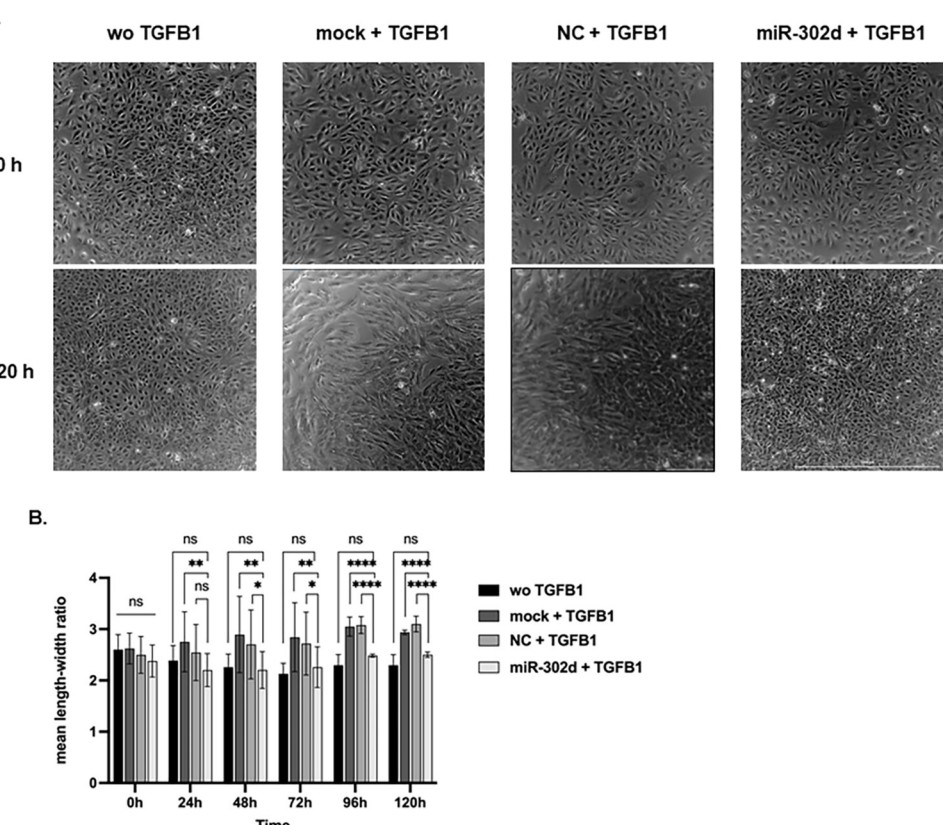

**Fig 2. miR-302d inhibits TGFB-induced fibroblastic morphological changes in hRPE cells.** hRPE cells were supplemented with or without TGFB1 for 5 d 24 h after transfection with mock, NC, or miR-302d. (A) The morphology of transfected hRPE cells at the beginning of TGFB1 exposure and 5 d later (S1 Video). The scale bar represents 1000 μm. (B) Mean length-width ratio changes of hRPE cells after TGFB1 exposure. Unpaired two-tailed t-tests (n = 3) were performed. The error bars represent SD. ns, non-significant, $^*p < 0.05$, $^{**}p < 0.01$, $^{****}p < 0.0001$.

suppress the increased production of FN1 protein under TGFB1 exposure ($p = 0.0198$ compared with "mock + TGFB1" group, $p = 0.0294$ compared with "NC + TGFB1" group, Fig 4).

We performed a wound-healing assay to examine cell motility, another important EMT feature. TGFB1 treatment enhanced the wound closure speed significantly after 24 h, reaching confluency around 84% at 72 h. By contrast, the migratory ability was significantly suppressed (around 56% confluency at 72 h, $p = 0.0359$) with the expression of miR-302d, comparable to that in cells without TGFB treatment (Fig 5).

## miR-302d reverts TGFB-induced mesenchymal hRPE cells toward an epithelial state

To investigate whether miR-302d can reverse TGFB-induced EMT, hRPE cells were pretreated with TGFB1 for four days and then transfected with mock, NC, miR-302d, or the TGFBR1 inhibitor SB431542 for 3 d. Phase-contrast pictures were recorded, and the mean length-width ratio was calculated to show the transformation of hRPE cells. 72 h later, hRPE cells with control transfections remained elongated, and the mean length-width ratio was slightly reduced due to proliferation. In contrast, cells transfected with miR-302d shifted to an epithelial and rounded morphology with a significantly reduced length-width ratio from 3.8 to 2.9, more evident than SB431542 ($p < 0.0001$, Fig 6).

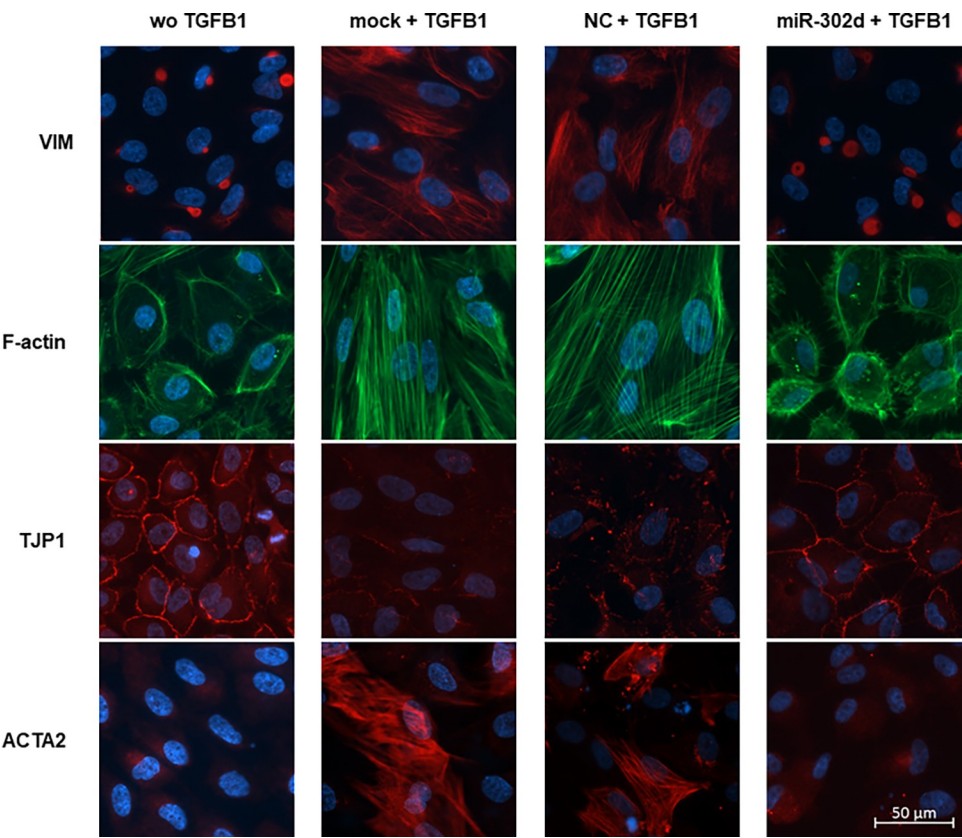

**Fig 3. miR-302d inhibits TGFB-induced changes of EMT-related factors in hRPE cells.** hRPE cells were treated with or without TGFB1 24 h after transfection with mock, NC, or miR-302d. Cells were immunostained for VIM, F-actin, TJP1, and ACTA2 4 d later. The scale bar represents 50 μm.

The expressions of EMT-related factors were visualized by ICC staining. With the transfection of miR-302d or SB431542 exposure, VIM and F-actin in cells shifted from a mesenchymal state to an epithelial distribution, and the immunofluorescence of αSMA was strongly attenuated (Fig 7). A significant reduction in FN1 protein level was also achieved by miR-302d in both ICC staining and Western blot ($p < 0.0001$ compared with "TGFB1 + mock" group, $p = 0.0085$ compared with "TGFB1 + NC" group, Fig 8).

## miR-302d suppresses TGFB-induced phosphorylation of SMAD2 in hRPE cells

In our previous study, SMAD2 and TGFBR2 were confirmed to be target genes of miR-302d and miR-302d prevented TGFB-induced phosphorylation of SMAD2 in ARPE-19 cells [6]. For verification in native hRPE cells, ICC staining detecting p-SMAD2 (pSMAD2) was performed 30, 60, and 90 min after TGFB1 stimulation 24 h post-transfection. For all time points, pSMAD2 was hardly detectable in the cells without TGFB1, whereas distinct elevation and nuclear translocation of pSMAD2 in both control groups were observed upon TGFB1 exposure. On the contrary, miR-302d inhibited the phosphorylation and nuclear localization of pSMAD2. As confirmed by Western blot, the pSMAD2 expression was reduced by around 64%, 73%, and 82% after 30, 60, and 90 min of TGFB1 exposure ($p = 0.004$ at 30 min, Fig 9, S1 and S2 Figs).

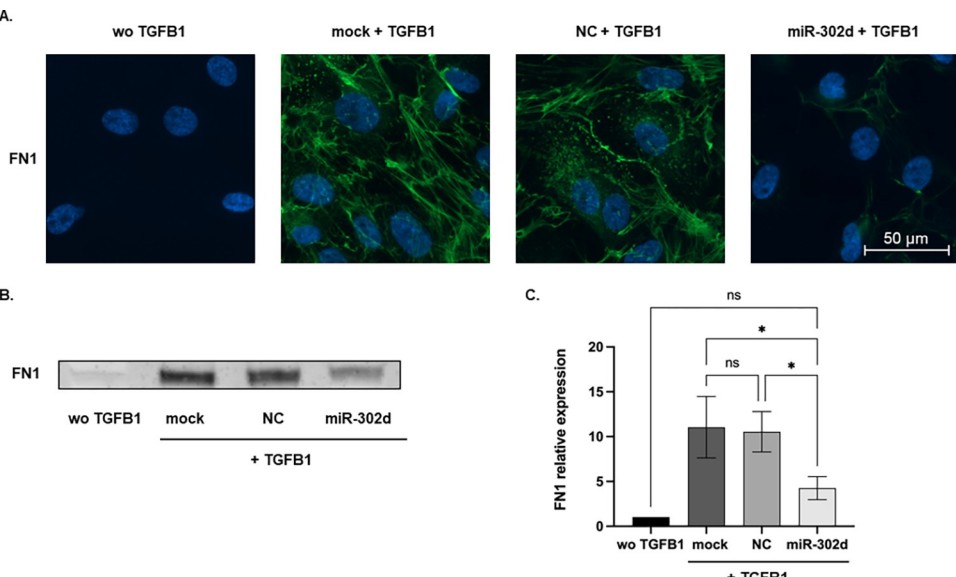

**Fig 4. miR-302d suppresses TGFB-induced production of FN1.** (A) ICC staining and (B) representative Western blot of FN1 in transfected hRPE cells after 3 d incubation of TGFB1. The scale bar represents 50 μm. (C) The relative expression of FN1 normalized to total proteins (S1 Raw images). The values were further normalized to the "wo TGFB1" group. One-way ANOVA with Tukey's multiple comparisons tests (n = 3) were performed. The error bars represent SD. ns, non-significant, $^*p < 0.05$.

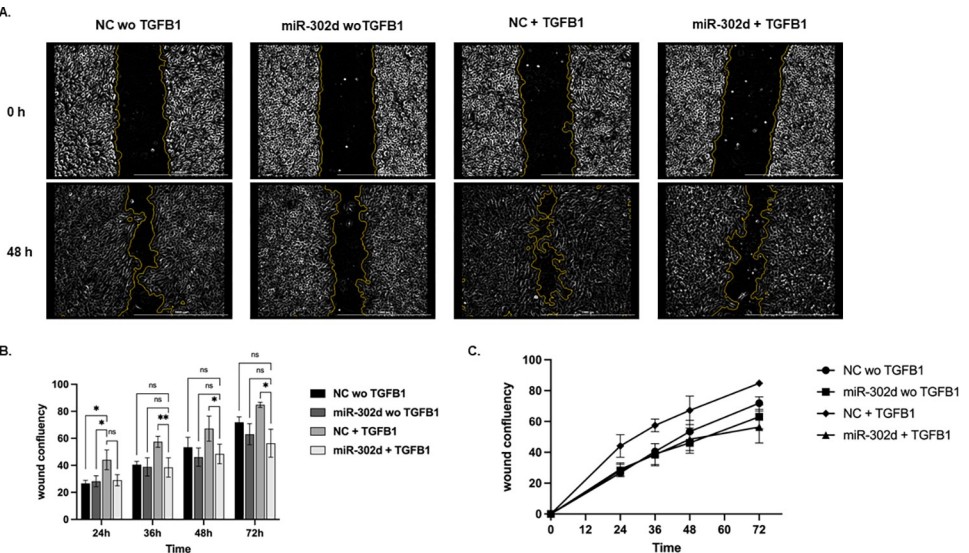

**Fig 5. miR-302d suppresses TGFB-induced migration in hRPE cells.** A scratch was generated in the monolayer of hRPE cells transfected with miR-302d or NC using a 200-μl pipette tip. Serum-free medium with or without TGFB1 was added and monitored for 72 h. (A) Representative scratch image after background subtraction with Gen5 software at the beginning and 48 h. The scale bar represents 1000 μm. (B) (C) The wound confluency over time. Two-way ANOVA with Fisher's LSD tests (n = 3) were performed. The error bars represent SD. ns, non-significant, $^*p < 0.05$, $^{**}p < 0.01$.

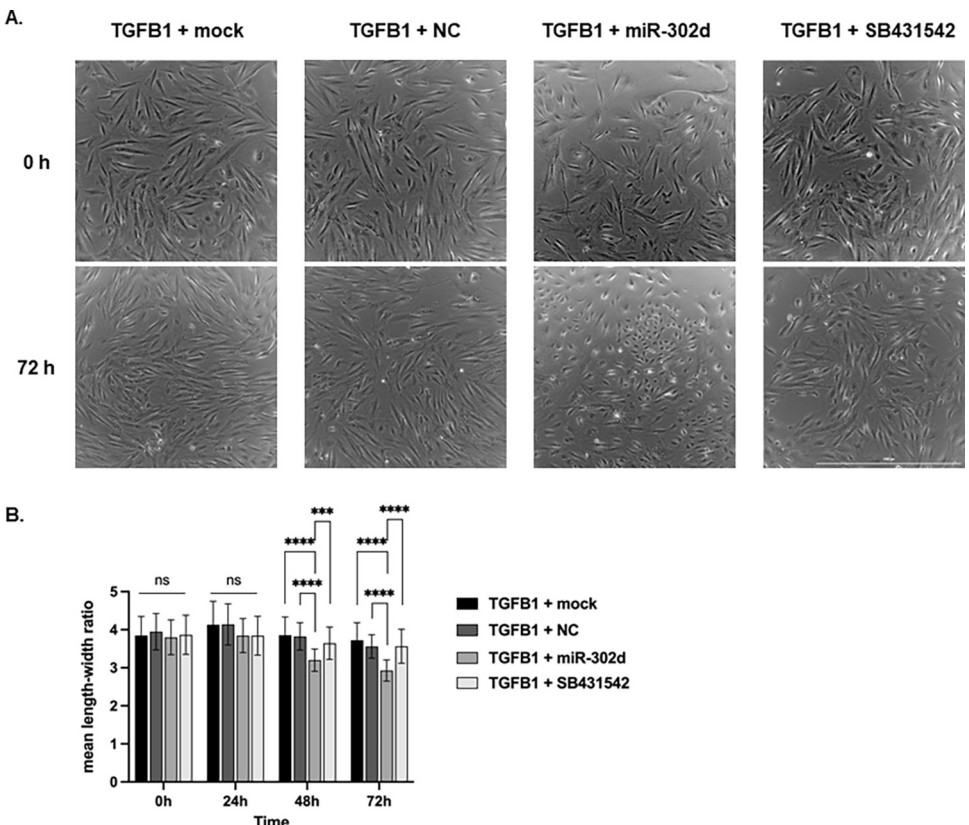

**Fig 6. miR-302d reverts TGFB-induced mesenchymal hRPE cells towards an epithelial morphology.** Following 4 d incubation of TGFB1, hRPE cells were transfected with mock, NC, miR-302d, or exposed to SB431542. (A) The morphology of hRPE cells at the beginning of transfection and 3 d later (S2 Video). The scale bar represents 1000 μm. (B) The mean length-width ratio changes of hRPE cells after transfection. Unpaired two-tailed t-tests (n = 3) were performed. The error bars represent SD. ns, non-significant, $^{***}p < 0.001$, $^{****}p < 0.0001$.

## miR-302d downregulates TGFBR2 and constitutive secretion of VEGFA in hRPE cells

As mentioned above, TGFBR2 and VEGFA were confirmed as target genes of miR-302d [6]. To verify the inhibition of TGFBR2 protein by miR-302d in hRPE cells, cell lysates were collected 48 and 72 hours after transfection, and TGFBR2 protein was measured by ELISA. Compared to the NC group, the TGFBR2 level in miR-302d transfected cells declined approximately 2-fold ($p < 0.0001$ at 48 h, $p = 0.0102$ at 72 h, Fig 10A).

In addition, the media were also collected and detected to quantify the concentration of VEGFA. Compared to NC, VEGFA expression declined around 20–30% with miR-302d ($p < 0.0001$ at 48 h, $p = 0.0015$ at 72 h, Fig 10B). The result suggested that miR-302d could reduce the endogenous secretion of VEGFA from hRPE cells.

## Discussion

EMT is a process by which polarized epithelial cells acquire mesenchymal properties through increased motility and invasiveness, cytoplasmic restructuring, overproduction of ECM components, and reduced matrix degradation [8]. It is a physiological process in normal embryogenesis and organ development [8, 26, 27]. However, it can be anomalously evoked by multiple factors such as inflammation, wounding, and cytokines and finally involved in

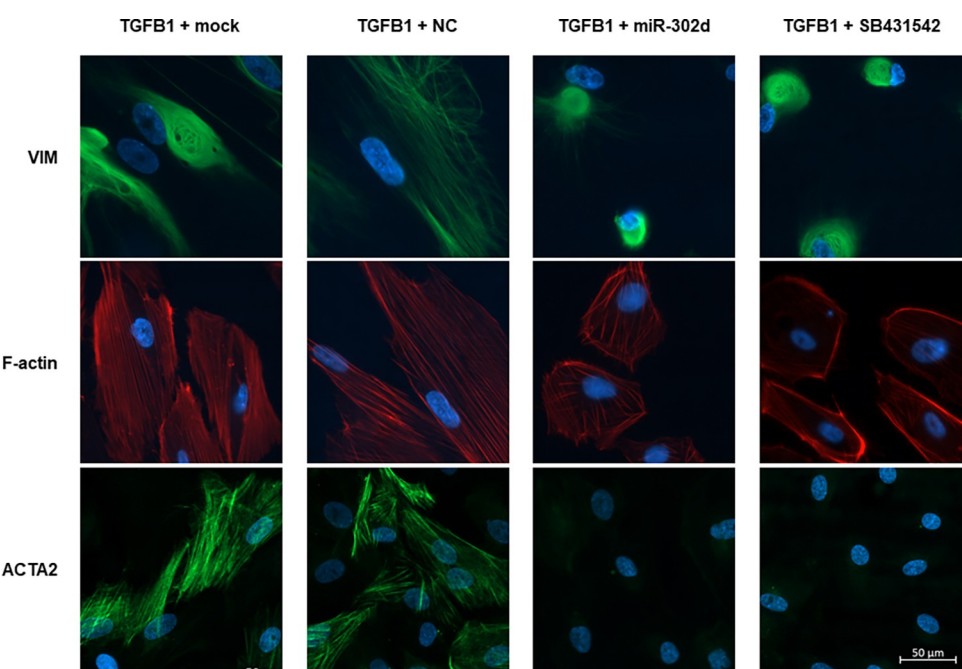

**Fig 7. miR-302d reverts TGFB-induced EMT in hRPE cells.** hRPE cells were incubated with TGFB1 for 4 d to achieve a mesenchymal state. Cells were then transfected with mock, NC, miR-302d, or exposed to SB431542. ICC staining for VIM, F-actin, and ACTA2 was performed 3 d post-transfection. The scale bar represents 50 μm.

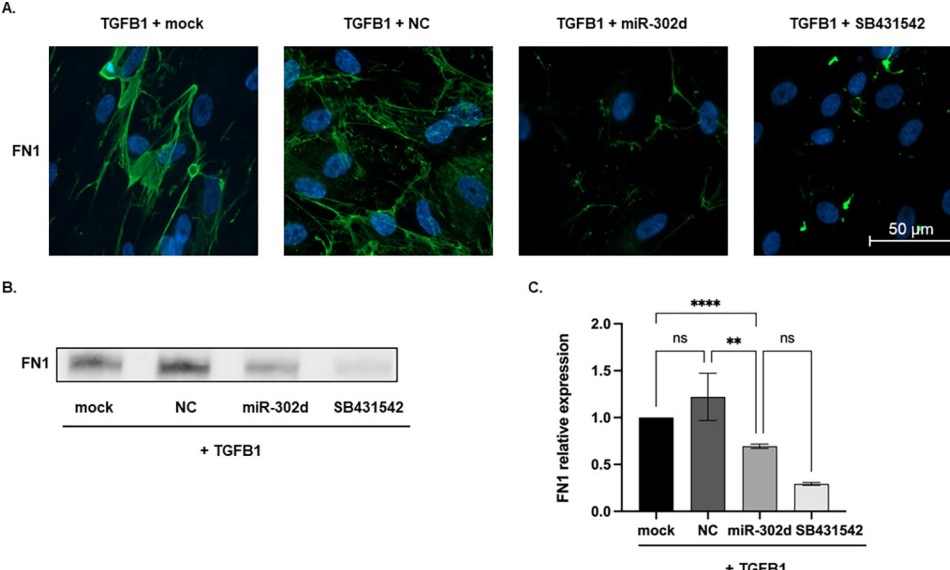

**Fig 8. miR-302d inhibits the FN1 level in mesenchymal hRPE cells.** (A) ICC staining and (B) representative Western blot of FN1 in hRPE cells after 4 d TGFB1 treatment and 3 d transfection. The scale bar represents 50 μm. (C) The relative expression of FN1 normalized to total proteins (S1 Raw images). The values were further normalized to the "TGFB1 + mock" group. One-way ANOVA with Tukey's multiple comparisons tests (n = 3) were performed. The error bars represent SD. ns, non-significant, $^{**}p < 0.01$, $^{****}p < 0.0001$.

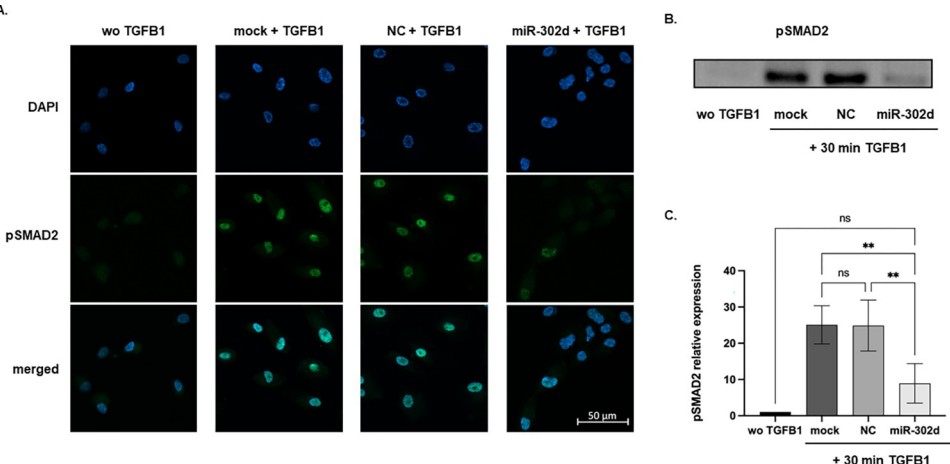

**Fig 9. miR-302d suppresses TGFB-induced phosphorylation of SMAD2 in hRPE cells.** hRPE cells were treated with or without TGFB1 for 30 min 24 h after transfection with mock, NC, or miR-302d. (A) DAPI (blue) and pSMAD2 (green) staining in hRPE cells. The scale bar represents 50 μm. (B) Representative Western blot of pSMAD2 30 min after initial TGFB1 exposure. (C) The relative expression of pSMAD2 normalized to total proteins (S1 Raw images). The values were further normalized to the "wo TGFB1" group. One-way ANOVA with Tukey's multiple comparisons tests (n = 3) were performed. The error bars represent SD. ns, non-significant, **$p < 0.01$.

pathological conditions, especially fibrosis and cancer [7, 27]. In fibrotic models of liver, lung, and kidney, which are primarily investigated, hepatocytes, alveolar and tubular epithelial cells transform into myofibroblasts [28–31]. In ocular diseases like corneal opacification, glaucoma, posterior capsular opacification, PVR, wAMD, and orbital fibrosis, transdifferentiation of myofibroblasts is also reported [4]. As an aberrant wound healing outcome of various posterior eye diseases, retinal fibrosis can disrupt normal RPE, photoreceptors, and vascular structures, causing permanent vision loss and treatment failures [5].

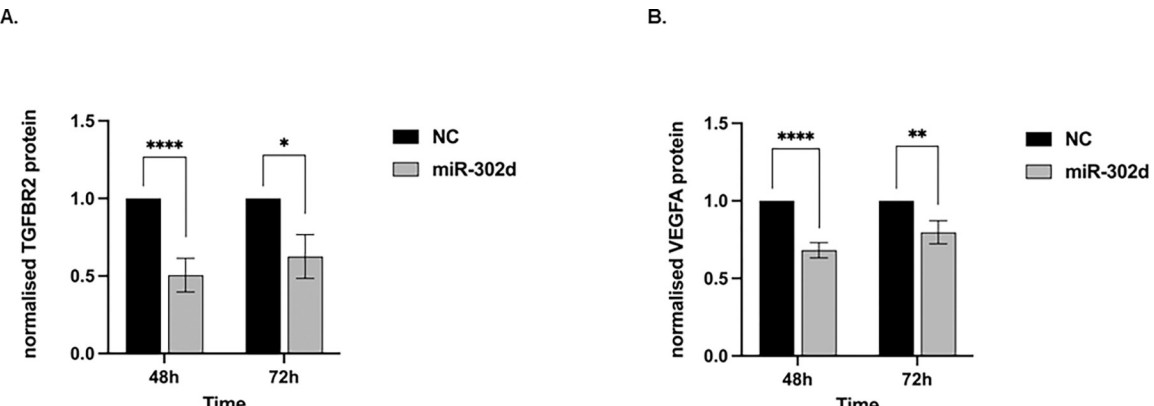

**Fig 10. miR-302d downregulates TGFBR2 and constitutive secretion of VEGFA in hRPE cells.** (A) ELISA for TGFBR2 protein levels 48 and 72 h after transfection with miR-302d relative to NC. The TGFBR2 protein was related to the total protein amounts determined by BCA assay. The amounts at 48 h were 10.35 ± 6.09 pg/mg in the NC group and 5.44 ± 3.49 pg/mg in the miR-302d group. The amounts at 72 h were 10.79 ± 4.31 in the NC group and 6.92 ± 3.45 in the miR-302d group. The values were further normalized to the NC group. (B) ELISA for secreted VEGFA protein levels 48 and 72 h after transfection with miR-302d relative to NC. The concentrations at 48 h were 423.93 ± 26.18 pg/ml in the NC group and 288.79 ± 26.18 pg/ml in the miR-302d group. The concentrations at 72 h were 658.87 ± 48.11 pg/ml in the NC group and 523.93 ± 44.09 pg/ml in the miR-302d group. The values were further normalized to the NC group. Unpaired two-tailed t-tests (n = 3) were performed. The error bars represent SD. ns, non-significant, *$p < 0.05$, **$p < 0.01$, ****$p < 0.0001$.

Numerous factors are reported to be capable of driving EMT, such as hepatocyte growth factor (HGF), insulin-like growth factor-1 (IGF1), epidermal growth factor (EGF), platelet-derived growth factor (PDGF), and connective tissues growth factor (CTGF) [12, 13]. The TGFB family is reported as a primary inducer [12, 13, 27, 32]. It is found to cause increased myofibroblast-like epithelial cells in the lung, kidney, and breast, and it correlates with the above alterations in fibroproliferative diseases in the eye [4, 27, 32, 33]. Of different TGFB isoforms, TGFB1 and TGFB2 are the major ones in RPE. While nontransformed RPE cells secrete TGFB2 predominantly, transformed RPE cells express higher levels of TGFB1 [34]. In addition, TGFB1 promotes the formation of epiretinal membranes and PVR and induces a significant mesenchymal transition in ARPE-19 cells starting from a dose of 10 ng/ml [3, 6, 35–37]. Therefore, this study used 10–20 ng/ml TGFB1 to stimulate transition in primary hRPE cells.

Recent non-surgical methods for retinal fibrosis treatment focus on administering antimetabolic drugs and blocking EMT by targeting the TGFB signaling pathway [32, 33]. EMT regulation is a complicated mechanism involving multiple genes, including epigenetic regulation, transcriptional control, and alternative splicing. Therefore, modulating EMT at the post-transcriptional level might be a promising approach [13, 16, 26]. With the function of regulating expressions of numerous genes, miRNAs as a therapeutic target or biomarker have become the focus of ophthalmologic research. For example, an increased level of miR-148 in the vitreous fluid seems to be associated with the severity of retinal detachment [38]. MiR-124 regulated EMT in ARPE-19 cells by TGFB/RHOG signaling pathway [36]. In our previous study, we screened miR-302d against multiple genes involved in TGFB signaling [6]. Here, we examined the effect of miR-302d on TGFB-induced changes in primary hRPE cells (Fig 11).

The transient expression of miR-302d repressed TGFB-induced EMT by keeping cells with a cobble-stone morphology with a stable length-width ratio and preventing cytoskeletal rearrangement, migration, and mesenchymal factors depositions. Targeting TGFBR2 and SMAD2 phosphorylation by miR-302d was confirmed with ELISA, ICC staining, and Western blot.

After exposure to TGFB1, hRPE cells underwent a mesenchymal transition. Transient expression of miR-302d restored the cells to their epithelial phenotype. This further demonstrated the therapeutic value of miR-302d by promoting MET.

Another non-negligible and critical factor elevated in the pathological process is VEGFA, which can be stimulated by TGFB through different signaling pathways in RPE cells [39]. VEGFA is a well-known pro-angiogenic element and contributes to retinal angiogenesis. In wAMD, ruptured blood-retinal barrier and leaky neovasculature lead to an aberrant microenvironment aggregated with inflammatory cells, immune cells, growth factors, and cytokines,

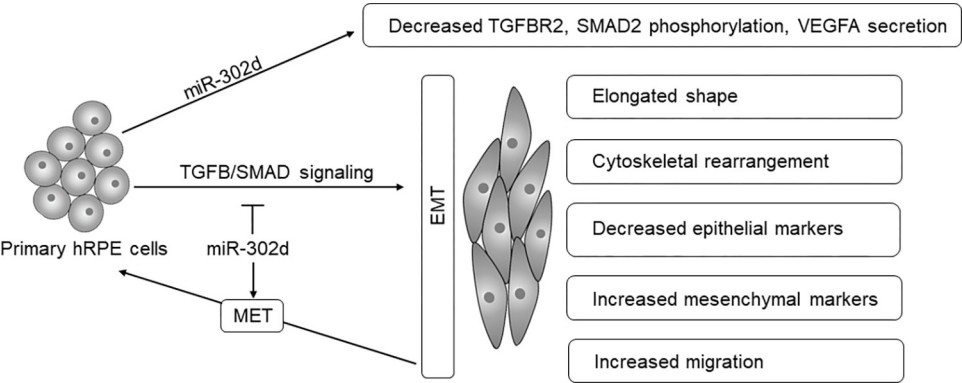

**Fig 11. An overall model depicts the effect of miR-302d on TGFB-induced EMT in hRPE cells.**

resulting in a fibrovascular lesion with eventual fibrosis [4, 40–42]. Our ELISA showed a statistically significant decline in VEGFA secretion 48 h after cells were transfected with miR-302d, which indicated that miR-302d could reduce the constitutive release of VEGFA from hRPE cells.

Although TGFB contributes significantly to inflammation suppression and ocular immune privilege, it may play a different role in the development of retinal fibrosis [43]. Histological studies reveal that immune elements, including macrophages, lymphocytes, and complement proteins, infiltrate retinal fibrosis [40]. Apart from transforming local cells in the retina, like endothelial cells and RPE cells, TGFB promotes the conversion of macrophages into myofibroblasts [12, 44]. Further research is required to understand the role of TGFB in the immune response to retinal fibrosis.

Taken together, miR-302d can inhibit TGFB-induced EMT and constitutive VEGFA secretion in primary hRPE cells by directly targeting TGFBR2 and disrupting the canonical TGFB/SMAD signaling cascade. Furthermore, it can promote MET to revert TGFB-induced fibroblast-like hRPE cells to an epithelial state. These findings support miR-302d as a putative therapeutic approach to (i) inhibit EMT, (ii) reverse proliferative retinal diseases, and (iii) assist in controlling angiogenesis in ocular neovascular disorders. Its dual function on fibrosis and neovascularization makes it a feasible option, especially for CNV-related diseases.

## Supporting information

**S1 Raw images.**
(PDF)

**S1 Fig. miR-302d suppresses TGFB-induced phosphorylation of SMAD2 in hRPE cells.**
hRPE cells were treated with or without TGFB1 for 60 min 24 h after transfection with mock, NC, or miR-302d. (A) DAPI (blue) and pSMAD2 (green) staining in hRPE cells. The scale bar represents 50 μm. (B) Representative Western blot of pSMAD2 60 min after initial TGFB1 exposure. (C) The relative expression of pSMAD2 normalized to total proteins (S1 Raw images). The values were further normalized to the "wo TGFB1" group.
(TIF)

**S2 Fig. miR-302d suppresses TGFB-induced phosphorylation of SMAD2 in hRPE cells.**
hRPE cells were treated with or without TGFB1 for 90 min 24 h after transfection with mock, NC, or miR-302d. (A) DAPI (blue) and pSMAD2 (green) staining in hRPE cells. The scale bar represents 50 μm. (B) Representative Western blot of pSMAD2 90 min after initial TGFB1 exposure. (C) The relative expression of pSMAD2 normalized to total proteins (S1 Raw images). The values were further normalized to the "wo TGFB1" group.
(TIF)

**S1 Video. A time-lapse movie of dynamic morphological change of hRPE cells after TGFB1 exposure 24 h post-transfection with NC or miR-302d.**
(MP4)

**S2 Video. A time-lapse movie of dynamic morphological change of hRPE cells after transfection with NC or miR-302d following 4 d TGFB1 exposure.**
(MP4)

## Author Contributions

**Conceptualization:** Heiko Fuchs.

**Data curation:** Xiaonan Hu.

**Formal analysis:** Xiaonan Hu, Heiko Fuchs.

**Investigation:** Xiaonan Hu.

**Methodology:** Heiko Fuchs.

**Project administration:** Heiko Fuchs.

**Resources:** Maximilian Binter, Karsten Hufendiek, Jan Tode, Carsten Framme.

**Supervision:** Carsten Framme, Heiko Fuchs.

**Visualization:** Xiaonan Hu, Heiko Fuchs.

**Writing – original draft:** Xiaonan Hu.

**Writing – review & editing:** Maximilian Binter, Karsten Hufendiek, Jan Tode, Carsten Framme, Heiko Fuchs.

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
