## [Decision Letter · Decision Letter 0]

2 Aug 2022

PONE-D-22-20602MiR-302d inhibits TGFB-induced EMT and promotes MET in primary human RPE cellsPLOS ONE

Dear Dr. Fuchs,

Thank you for submitting your manuscript to PLOS ONE. After careful consideration, we feel that it has merit but does not fully meet PLOS ONE’s publication criteria as it currently stands. Therefore, we invite you to submit a revised version of the manuscript that addresses the points raised during the review process.

Both reviewers see significant impact for the RPE research in your paper. There are a couple of changes and improvements required. mostly at the level methods description and performance. Some RPE markers in the IHC would useful. Please submit your revised manuscript by Sep 16 2022 11:59PM. If you will need more time than this to complete your revisions, please reply to this message or contact the journal office at plosone@plos.org. Please include the following items when submitting your revised manuscript:A rebuttal letter that responds to each point raised by the academic editor and reviewer(s). You should upload this letter as a separate file labeled 'Response to Reviewers'.A marked-up copy of your manuscript that highlights changes made to the original version. You should upload this as a separate file labeled 'Revised Manuscript with Track Changes'.An unmarked version of your revised paper without tracked changes. You should upload this as a separate file labeled 'Manuscript'.

We look forward to receiving your revised manuscript.

Kind regards,

Olaf Strauß

Academic Editor

PLOS ONE

Journal Requirements:

"The authors received no specific funding for this work."

Reviewers' comments:

Reviewer's Responses to Questions

**Comments to the Author**

1. Is the manuscript technically sound, and do the data support the conclusions?

Reviewer #1: Partly

Reviewer #2: Yes

2. Has the statistical analysis been performed appropriately and rigorously? 

Reviewer #1: Yes

Reviewer #2: Yes

3. Have the authors made all data underlying the findings in their manuscript fully available?

Reviewer #1: Yes

Reviewer #2: Yes

4. Is the manuscript presented in an intelligible fashion and written in standard English?

Reviewer #1: Yes

Reviewer #2: Yes

5. Review Comments to the Author

Reviewer #1: In this manuscript, Hu et al., demonstrated that miR-302d inhibits TGFB-induced epithelial–mesenchymal transition in primary human retinal pigment epithelial cells by targeting TGFBR2. Below are my major comments,

1. In abstract: The authors have directly mentioned the purpose of the study which is to assess whether microRNA (miR-302d) inhibits EMT in primary human RPE cells without giving a brief rationale as why have they selected miR-302d.

2. Authors should address what is absolute amount of miR-302d in primary human RPE cells. Or at least, author should demonstrate relative abundance of miR-302d in cells compared to other well-known microRNAs by qPCR or analyzing publicly available NGS data in primary human RPE cells.

3. As we known, many inflammatory cytokines, such as TGF-β and TNF-α, act as potent inducers of the EMT in RPE cells, which is implicated in the pathogenesis of PVR. TGF-β is known to have three isoforms: TGF-β1, TGF-β2, and TGF-β3. Authors should address the rationale about why this study used TGF-β1 instead of other isoforms in the subsequent experiments.

4. RPE cells activated by growth factors and cytokines subsequently undergo EMT, and the resulting dedifferentiated RPE cells migrate, proliferate, and transform into proliferative migratory spindle cells, resulting in PVR. Authors should address more about EMT features by using scratch-wound healing, modified Boyden chamber assay, or collagen gel contraction assay in this study.

5. Most of data authors observed are from the effects of miR-302d overexpression in RPE cells. However, as overexpressed microRNA could have significant off-target issue, which are not biology relevance. To solve this issue, at least, authors should treat the RPE cells with miR-302d inhibitor. This should be done across all the experiment design.

6. Authors should add a model in last figure depicting the take home message of this manuscript. Discuss the model in discussion part. It will be helpful for readers to understand the global message.

Reviewer #2: In their study, the authors show how the MiR-302d interferes with the mesenchymal transition of primary human RPE cells. The study is of great interest. I have some comments, mainly concerning the methods, which need to addressed.

Methods

Cell isolation and culture

Please give the approximate time between enucleation and RPE preparation. Cultures from how many patients were obtained and how much passages (if any) were conducted with each culture? Were their any exclusion criteria? Why were the eyes enucleated? RPE sheet, were this exclusively RPE cells or RPE/choroid tissue (as suggested by the description in the method section).

Line 125/126, I am not sure I understand the sentence. Was the same RPE sheet digested for three times in total? If so, were the cells pooled? In what kind of plate were the cells seeded and how long were their cultured before experimentation? How was the quality and morphology/differentiation of the cells assessed? If RPE/choroid sheets were used, identity of the RPE should be verified. This is especially important if eyes of uveal melanoma patients were used.

If the cells were passaged, how was the passage conducted?

Line 131 “one day in advance”, one day before the transfection, please rephrase. Were these cells seeded directly from the preparation or from passaged cells, if so, which passage?

Transfection, how was the success of the transfection verified? Please elaborate

Please explain the concentration of TGFB used (reference?).

IHC, were any RPE marker included?

Western blot, please mention the time of treatment before harvest of the cells.

Western blot, please mention the wavelength of measurement and the software used for the calculation of the concentration.

ELiSA, please mention how long the supernatant was collected.

Statistical analysis, it is not clear how many independent experiments were conducted, please specify.

Results and figures, please give the sample size (n) for all experiments in the figure legends.

Line 310, please mention in text what SB431542 is inhibiting

Line 370, following, please give the mean and range of concentration you measured for VEGF and TGFBR2.

Discussion

Please discuss the results concerning the current literature, not just reiterate the findings.

RPE cells secrete quite high amounts of TGFß which are important for the immune privilege of the retina. Please discuss your findings also in relation to this.

6. PLOS authors have the option to publish the peer review history of their article (what does this mean?). If published, this will include your full peer review and any attached files.

Reviewer #1: No

Reviewer #2: No

---

## [Author Response · Author response to Decision Letter 0]

27 Sep 2022

Responses to the Reviewer #1 comments.

In this manuscript, Hu et al., demonstrated that miR-302d inhibits TGFB-induced epithelial–mesenchymal transition in primary human retinal pigment epithelial cells by targeting TGFBR2. Below are my major comments,

1. In abstract: The authors have directly mentioned the purpose of the study which is to assess whether microRNA (miR-302d) inhibits EMT in primary human RPE cells without giving a brief rationale as why have they selected miR-302d.

Answer: 

We revised the abstract (Line 19-22): “Our previous study identified the miRNA miR-302d as a regulator of multiple TGFB-related genes in ARPE-19 cells. To further explore its effect on primary cells, the effect of miR-302d on TGFB-induced EMT in primary human retinal pigment epithelium (hRPE) was investigated in vitro.”

2. Authors should address what is absolute amount of miR-302d in primary human RPE cells. Or at least, author should demonstrate relative abundance of miR-302d in cells compared to other well-known microRNAs by qPCR or analyzing publicly available NGS data in primary human RPE cells.

Answer: 

We added qPCR assay testing the amount of miR-302d and other miRNAs such as miR-17, miR-27a, miR-27b, which have been reported to be expressed in RPE cells and used RNU19 as a housekeeping gene. The result showed that hRPE cells do not endogenously express miR-302d (Line 250-254): “In addition, a TaqMan® MicroRNA assay was performed to determine whether hRPE cells endogenously express miR-302d. MiR-17, miR-27a, and miR-27b were used as controls, as their expressions were reported in RPE cells, and RNU19 was used as the housekeeping gene [24,25]. According to the results, hRPE cells do not endogenously express miR-302d (Fig 1B).”

3. As we known, many inflammatory cytokines, such as TGF-β and TNF-α, act as potent inducers of the EMT in RPE cells, which is implicated in the pathogenesis of PVR. TGF-β is known to have three isoforms: TGF-β1, TGF-β2, and TGF-β3. Authors should address the rationale about why this study used TGF-β1 instead of other isoforms in the subsequent experiments.

Answer:

The reason that TGFB1 was used is added in the Discussion section (Line 437-445): “The TGFB family is reported as a primary inducer [12,13,27,32]. It is found to cause increased myofibroblast-like epithelial cells in the lung, kidney, and breast, and it correlates with the above alterations in fibroproliferative diseases in the eye [4,27,32,33]. Of different TGFB isoforms, TGFB1 and TGFB2 are the major ones in RPE. While nontransformed RPE cells secrete TGFB2 predominantly, transformed RPE cells express higher levels of TGFB1 [34]. In addition, TGFB1 promotes the formation of epiretinal membranes and PVR and induces a significant mesenchymal transition in ARPE-19 cells starting from a dose of 10 ng/ml [3,6,35–37]. Therefore, this study used 10-20 ng/ml TGFB1 to stimulate transition in primary hRPE cells.”

4. RPE cells activated by growth factors and cytokines subsequently undergo EMT, and the resulting dedifferentiated RPE cells migrate, proliferate, and transform into proliferative migratory spindle cells, resulting in PVR. Authors should address more about EMT features by using scratch-wound healing, modified Boyden chamber assay, or collagen gel contraction assay in this study.

Answer:

We added a wound healing assay to exhibit the feature of EMT and the function of miR-302d (Line 302-306 and Fig 5): “We performed a wound-healing assay to examine cell motility, another important EMT feature. TGFB1 treatment enhanced the wound closure speed significantly after 24 h, reaching confluency around 84% at 72 h. By contrast, the migratory ability was significantly suppressed (around 56% confluency at 72 h, p = 0.0359) with the expression of miR-302d, comparable to that in cells without TGFB treatment (Fig 5).”

5. Most of data authors observed are from the effects of miR-302d overexpression in RPE cells. However, as overexpressed microRNA could have significant off-target issue, which are not biology relevance. To solve this issue, at least, authors should treat the RPE cells with miR-302d inhibitor. This should be done across all the experiment design.

Answer:

We do not see any point in treating cells with a miR-302d inhibitor because of the following reasons: 

1. For transient miRNA overexpression, we always try to keep the applied miRNA concentration as low as possible in order not to hinder miRNA maturation of endogenous miRNAs, which could lead to off-target effects. To exclude this, we performed a scrambled negative control and a mock transfection for our initial experiments. Our experience has shown remarkable effects can also be achieved with a scrambled negative control transfection if too high concentrations are used, overloading the RNAi-induced silencing complex so endogenous miRNAs can no longer become active.

2. According to the literature, miR-302d is activated together with four other microRNAs (mir-302a, miR-302b, miR-302c and mir-367) as a polycytronic cluster of pluripotency-associated transcription factors such as OCT4, SOX2 and Nanog. This microRNA cluster is thus expressed in embryonic stem cells and inhibits genes that initiate differentiation, such as TGFBR2, promoting stem cell commitment.

3. However, our TaqMan microRNA assay confirmed that hRPE cells do not endogenously express miR-302d, in contrast to miR-17, miR-27a or miR-27b, and we included these data in Fig 1.

6. Authors should add a model in last figure depicting the take home message of this manuscript. Discuss the model in discussion part. It will be helpful for readers to understand the global message.

Answer:

We added a model in the last figure discussed in the Discussion section (Fig 11, Line 487-494): “Taken together, miR-302d can inhibit TGFB-induced EMT and constitutive VEGFA secretion in primary hRPE cells by directly targeting TGFBR2 and disrupting the canonical TGFB/SMAD signaling cascade. Furthermore, it can promote MET to revert TGFB-induced fibroblast-like hRPE cells to an epithelial state. These findings support miR-302d as a putative therapeutic approach to (i) inhibit EMT, (ii) reverse proliferative retinal diseases, and (iii) assist in controlling angiogenesis in ocular neovascular disorders. Its dual function on fibrosis and neovascularization makes it a feasible option, especially for CNV-related diseases.”

Responses to the Reviewer #2 comments

In their study, the authors show how the MiR-302d interferes with the mesenchymal transition of primary human RPE cells. The study is of great interest. I have some comments, mainly concerning the methods, which need to addressed.

Methods

Cell isolation and culture

Please give the approximate time between enucleation and RPE preparation. Cultures from how many patients were obtained and how much passages (if any) were conducted with each culture? Were there any exclusion criteria? Why were the eyes enucleated? RPE sheet, were this exclusively RPE cells or RPE/choroid tissue (as suggested by the description in the method section).

Answer: 

It was only the RPE sheet that was peeled off for cell culture as suggested in the manuscript. The details of cell isolation and culture, including selection criteria of patients were added in the Methods section (Line 93-97): “Primary hRPE cells were obtained from two female patients who received enucleations suffering from an early stage of painful Phthisis bulbi. Patients with previous eye disorders involving the retina were excluded. The eyeballs were packed in ROTI®Cell Hanks' BSS (HBSS, Carl Roth #9117.1) on ice and processed to harvest RPE cells within 1-2 hours after surgery.”

Line 125/126, I am not sure I understand the sentence. Was the same RPE sheet digested for three times in total? If so, were the cells pooled? In what kind of plate were the cells seeded and how long were their cultured before experimentation? How was the quality and morphology/differentiation of the cells assessed? If RPE/choroid sheets were used, identity of the RPE should be verified. This is especially important if eyes of uveal melanoma patients were used.

Answer: 

1. We modified the Methods and Results sections in response to this comment (line 102-118), stating: “The RPE sheet was peeled off with forceps and then digested with 1 ml TrypLE™ Express Enzyme (Gibco #12604-021) on a thermo-shaker for 30 min, 37°C, and 600 rpm. The detached RPE cells were collected by centrifugation and then resuspended and cultured in a 6-well plate with Minimum Essential Medium Eagle media (MEM, Sigma-Aldrich #M8042) supplemented with 10% fetal bovine serum (FBS, Pan-Biotech #P40-39500), 1% GlutaMAX™ (Gibco #3505-061), and 1% Penicillin-Streptomycin (Pen-Strep, Gibco #15140-122). The same RPE sheet was digested two more times for 30 min each, and the RPE cells were transferred to different wells. Sequential digestion was performed to avoid over digestion of already detached cells, resulting in poor attachment. When the cells were confluent, they were examined with a phase-contrast and brightfield microscope (Leica DMi1) regarding their epithelial properties and degree of pigmentation. Wells showing RPE cells with similar morphology were pooled in passage 1 with 400 µl TrypLE™, and cells of passages 4-5 were used for experiments. It should be mentioned that RPE cells seeded at low density changed to a mesenchymal state. Therefore, the cells were seeded at a density of 50% to maintain their epithelial state, i.e., the cells of one confluent 6-well were divided into two 6-wells.”

2. The identification of extracted hRPE cells was confirmed by ICC staining of its classic marker, RPE65, and its phagocytic function. The Results section elucidated the results (Line 243-249), stating: “In order to identify hRPE cells, some specific characteristics were examined. ICC staining revealed abundant expression of RPE65, a native marker of RPE cells, in extracted cells. Moreover, fluorescent microbeads were added into the medium for four days to test the phagocytic property of RPE cells. The green fluorescent microbeads were visible within the cytoplasm, which was confirmed by the coexistence with Filamentous actin (F-actin). The findings validated that primary hRPE cells were successfully extracted (Fig 1A).”.

If the cells were passaged, how was the passage conducted?

Answer: 

We revised the Methods section in response to this comment (Line 114-118), stating: “Wells showing RPE cells with similar morphology were pooled in passage 1 with 400 µl TrypLE™, and cells of passages 4-5 were used for experiments. It should be mentioned that RPE cells seeded at low density changed to a mesenchymal state. Therefore, the cells were seeded at a density of 50% to maintain their epithelial state, i.e., the cells of one confluent 6-well were divided into two 6-wells.”

Line 131 “one day in advance”, one day before the transfection, please rephrase. Were these cells seeded directly from the preparation or from passaged cells, if so, which passage?

Answer: 

1. We rephrased the text to stress that (Line 139-140), stating: “2 x 104 primary hRPE cells were seeded in each well of a 24-well plate with a complete medium to reach 70-80% confluency one day before the transfection.”

2. The cells of passages 4-5 were used for experiments, which was added in the Methods section (Line 114-116), stating: “Wells showing RPE cells with similar morphology were pooled in passage 1 with 400 µl TrypLE™, and cells of passages 4-5 were used for experiments.”

Transfection, how was the success of the transfection verified? Please elaborate.

Answer:

The added TaqMan microRNA assay showed that hRPE cells do not endogenously express miR-302d. Therefore, the comparison between miR-302d-transfected cells and negative control-transfected cells in our results could prove the success of the transfection. 

Please explain the concentration of TGFB used (reference?).

Answer:

The reason of the concentration we used was added in the Discussion section (Line 440-445), stating: “Of different TGFB isoforms, TGFB1 and TGFB2 are the major ones in RPE. While nontransformed RPE cells secrete TGFB2 predominantly, transformed RPE cells express higher levels of TGFB1 [34]. In addition, TGFB1 promotes the formation of epiretinal membranes and PVR and induces a significant mesenchymal transition in ARPE-19 cells starting from a dose of 10 ng/ml [3,6,35–37]. Therefore, this study used 10-20 ng/ml TGFB1 to stimulate transition in primary hRPE cells.” 

IHC, were any RPE marker included?

Answer:

ICC staining of the classic RPE marker, RPE65, and its phagocytic function was added. The results were elucidated in the Results section (Line 243-249), stating: “In order to identify hRPE cells, some specific characteristics were examined. ICC staining revealed abundant expression of RPE65, a native marker of RPE cells, in extracted cells. Moreover, fluorescent microbeads were added into the medium for four days to test the phagocytic property of RPE cells. The green fluorescent microbeads were visible within the cytoplasm, which was confirmed by the coexistence with Filamentous actin (F-actin). The findings validated that primary hRPE cells were successfully extracted (Fig 1A).”

Western blot, please mention the time of treatment before harvest of the cells.

Answer: 

The respective time was added in the figure legends (Line 312-314, 360-362 and 385-386), stating: “(A) ICC staining and (B) representative Western blot of FN1 in transfected hRPE cells after 3 d incubation of TGFB1.”, “(A) ICC staining and (B) representative Western blot of FN1 in hRPE cells after 4 d TGFB1 treatment and 3 d transfection.”, and “(B) Representative Western blot of pSMAD2 30 min after initial TGFB1 exposure.”

Western blot, please mention the wavelength of measurement and the software used for the calculation of the concentration.

Answer:

1. We revised the Methods section in response to this comment (Line 187-191), stating: “After incubation with 1:1000 diluted goat anti-rabbit secondary antibody StarBright™ Blue 700 (Bio-Rad #12004162) at RT for one hour, the fluorescent signal was detected with ChemiDoc MP Imaging System (Bio-Rad) at a measurement wavelength of 660-720 nm. ImageLab 6 software (Bio-Rad) was used for normalization and quantification.”

2. The data were analyzed using Microsoft Excel 2010 and GraphPad Prism 9, as mentioned in the Statistical analysis section (Line 235-239), stating: “All experiments were performed in three independent replicates, and the data were analyzed using Microsoft Excel 2010 and GraphPad Prism 9. Differences were evaluated by unpaired two-tailed t-test, one-way ANOVA with Tukey's multiple comparisons test or two-way ANOVA with Fisher's LSD test as the post hoc test. A p value < 0.05 was considered significantly different.”

ELiSA, please mention how long the supernatant was collected.

Answer:

This was mentioned in the figure legend of Fig 10B (Line 406-407), stating: “(B) ELISA for secreted VEGFA protein levels 48 and 72 h after transfection with miR-302d relative to NC.”

Statistical analysis, it is not clear how many independent experiments were conducted, please specify.

Answer: 

We revised the text to stress that (Line 235), stating: “All experiments were performed in three independent replicates, and the data were analyzed using Microsoft Excel 2010 and GraphPad Prism 9.”

Results and figures, please give the sample size (n) for all experiments in the figure legends.

Answer: 

We added respective statistical methods and sample sizes in figure legends (Line 260-261, 279, 316-317, 324-325, 344, 364-365, 388-389, 411), stating: “One-sample t-tests (n=3) were performed to detect differences between mean Cq values and 35.”, “Unpaired two-tailed t-tests (n=3) were performed.”, “One-way ANOVA with Tukey's multiple comparisons tests (n=3) were performed.”, “Two-way ANOVA with Fisher's LSD tests (n=3) were performed.”, “Unpaired two-tailed t-tests (n=3) were performed.”, “One-way ANOVA with Tukey's multiple comparisons tests (n=3) were performed.”, “One-way ANOVA with Tukey's multiple comparisons tests (n=3) were performed.”, “Unpaired two-tailed t-tests (n=3) were performed.”.

Line 310, please mention in text what SB431542 is inhibiting

Answer:

We modified the text to that (Line 331) and mentioned it in the Methods section (Line 145), stating: “… the TGFBR1 inhibitor SB431542”.

Line 370, following, please give the mean and range of concentration you measured for VEGF and TGFBR2.

Answer: 

The ranges of concentrations for VEGF and TGFBR2 were added in figure legends (Line 402-410), stating: “The TGFBR2 protein was related to the total protein amounts determined by BCA assay. The amounts at 48 h were 10.35 ± 6.09 pg/mg in the NC group and 5.44 ± 3.49 pg/mg in the miR-302d group. The amounts at 72 h were 10.79 ± 4.31 in the NC group and 6.92 ± 3.45 in the miR-302d group. The values were further normalized to the NC group. (B) ELISA for secreted VEGFA protein levels 48 and 72 h after transfection with miR-302d relative to NC. The concentrations at 48 h were 423.93 ± 26.18 pg/ml in the NC group and 288.79 ± 26.18 pg/ml in the miR-302d group. The concentrations at 72 h were 658.87 ± 48.11 pg/ml in the NC group and 523.93 ± 44.09 pg/ml in the miR-302d group.”

Discussion

Please discuss the results concerning the current literature, not just reiterate the findings.

RPE cells secrete quite high amounts of TGFß which are important for the immune privilege of the retina. Please discuss your findings also in relation to this.

Answer:

We revised the Discussion section to make it more concise (451-486), stating: “With the function of regulating expressions of numerous genes, miRNAs as a therapeutic target or biomarker have become the focus of ophthalmologic research. For example, an increased level of miR-148 in the vitreous fluid seems to be associated with the severity of retinal detachment [38]. MiR-124 regulated EMT in ARPE-19 cells by TGFB/RHOG signaling pathway [36]. In our previous study, we screened miR-302d against multiple genes involved in TGFB signaling [6]. Here, we examined the effect of miR-302d on TGFB-induced changes in primary hRPE cells (Fig 11).

The transient expression of miR-302d repressed TGFB-induced EMT by keeping cells with a cobble-stone morphology with a stable length-width ratio and preventing cytoskeletal rearrangement, migration, and mesenchymal factors depositions. Targeting TGFBR2 and SMAD2 phosphorylation by miR-302d was confirmed with ELISA, ICC staining, and Western blot.

After exposure to TGFB1, hRPE cells underwent a mesenchymal transition. Transient expression of miR-302d restored the cells to their epithelial phenotype. This further demonstrated the therapeutic value of miR-302d by promoting MET.

Another non-negligible and critical factor elevated in the pathological process is VEGFA, which can be stimulated by TGFB through different signaling pathways in RPE cells [39]. VEGFA is a well-known pro-angiogenic element and contributes to retinal angiogenesis. In wAMD, ruptured blood-retinal barrier and leaky neovasculature lead to an aberrant microenvironment aggregated with inflammatory cells, immune cells, growth factors, and cytokines, resulting in a fibrovascular lesion with eventual fibrosis [4,40–42]. Our ELISA showed a statistically significant decline in VEGFA secretion 48 h after cells were transfected with miR-302d, which indicated that miR-302d could reduce the constitutive release of VEGFA from hRPE cells.

Although TGFB contributes significantly to inflammation suppression and ocular immune privilege, it may play a different role in the development of retinal fibrosis [43]. Histological studies reveal that immune elements, including macrophages, lymphocytes, and complement proteins, infiltrate retinal fibrosis [40]. Apart from transforming local cells in the retina, like endothelial cells and RPE cells, TGFB promotes the conversion of macrophages into myofibroblasts [12,44]. Further research is required to understand the role of TGFB in the immune response to retinal fibrosis.

---

## [Decision Letter · Decision Letter 1]

11 Nov 2022

MiR-302d inhibits TGFB-induced EMT and promotes MET in primary human RPE cells

PONE-D-22-20602R1

Dear Dr. Fuchs,

We’re pleased to inform you that your manuscript has been judged scientifically suitable for publication and will be formally accepted for publication once it meets all outstanding technical requirements.

Kind regards,

Olaf Strauß

Academic Editor

PLOS ONE

Additional Editor Comments (optional):

Reviewers' comments:

Reviewer's Responses to Questions

**Comments to the Author**

1. If the authors have adequately addressed your comments raised in a previous round of review and you feel that this manuscript is now acceptable for publication, you may indicate that here to bypass the “Comments to the Author” section, enter your conflict of interest statement in the “Confidential to Editor” section, and submit your "Accept" recommendation.

Reviewer #1: All comments have been addressed

Reviewer #2: All comments have been addressed

2. Is the manuscript technically sound, and do the data support the conclusions?

Reviewer #1: Partly

Reviewer #2: Yes

3. Has the statistical analysis been performed appropriately and rigorously? 

Reviewer #1: N/A

Reviewer #2: Yes

4. Have the authors made all data underlying the findings in their manuscript fully available?

Reviewer #1: Yes

Reviewer #2: Yes

5. Is the manuscript presented in an intelligible fashion and written in standard English?

Reviewer #1: Yes

Reviewer #2: Yes

6. Review Comments to the Author

Reviewer #1: Thank you for submitting your manuscript, which was very interesting and adequately describes the MiR-302d inhibits TGFB-induced EMT and promotes MET in primary human RPE cells. However, I felt there still lacks evidence in fulfilling the whole story, specifically animal studies. I believe after the author has completed the animal study, this manuscript will be complete.

Reviewer #2: (No Response)

7. PLOS authors have the option to publish the peer review history of their article (what does this mean?). If published, this will include your full peer review and any attached files.

Reviewer #1: No

Reviewer #2: No

---

## [Editor Report · Acceptance letter]

15 Nov 2022

PONE-D-22-20602R1 

MiR-302d inhibits TGFB-induced EMT and promotes MET in primary human RPE cells 

Dear Dr. Fuchs:

I'm pleased to inform you that your manuscript has been deemed suitable for publication in PLOS ONE. Congratulations! Your manuscript is now with our production department. 

Kind regards, 

on behalf of

Professor Olaf Strauß 

Academic Editor

PLOS ONE